# Progress in Nonalcoholic Fatty Liver Disease: SIRT Family Regulates Mitochondrial Biogenesis

**DOI:** 10.3390/biom12081079

**Published:** 2022-08-05

**Authors:** Chuanfei Zeng, Mingkai Chen

**Affiliations:** Department of Gastroenterology, Renmin Hospital of Wuhan University, No. 99 Zhang Zhidong Road, Wuhan 430060, China

**Keywords:** SIRT family, mitochondria, nonalcoholic fatty liver disease, research progress

## Abstract

Nonalcoholic fatty liver disease (NAFLD) is characterized by hepatic steatosis, insulin resistance, mitochondrial dysfunction, inflammation, and oxidative stress. As a group of NAD+-dependent III deacetylases, the sirtuin (SIRT1–7) family plays a very important role in regulating mitochondrial biogenesis and participates in the progress of NAFLD. SIRT family members are distributed in the nucleus, cytoplasm, and mitochondria; regulate hepatic fatty acid oxidation metabolism through different metabolic pathways and mechanisms; and participate in the regulation of mitochondrial energy metabolism. SIRT1 may improve NAFLD by regulating ROS, PGC-1α, SREBP-1c, FoxO1/3, STAT3, and AMPK to restore mitochondrial function and reduce steatosis of the liver. Other SIRT family members also play a role in regulating mitochondrial biogenesis, fatty acid oxidative metabolism, inflammation, and insulin resistance. Therefore, this paper comprehensively introduces the role of SIRT family in regulating mitochondrial biogenesis in the liver in NAFLD, aiming to further explain the importance of SIRT family in regulating mitochondrial function in the occurrence and development of NAFLD, and to provide ideas for the research and development of targeted drugs. Relatively speaking, the role of some SIRT family members in NAFLD is still insufficiently clear, and further research is needed.

## 1. Introduction

Nonalcoholic fatty liver disease (NAFLD) is the general term for a range of liver disease states of progressive severity, including simple steatosis, nonalcoholic steatohepatitis (NASH) with fibrosis, liver cirrhosis, and hepatocellular carcinoma [1]. A survey report showed that the number of NAFLD patients in the United States is expected to increase from 83.1 million in 2015 to 109 million in 2030, bringing a heavy economic burden as well as an increasing demand for liver transplantation and an increased risk of hepatocellular carcinoma in patients with liver cirrhosis and end-stage liver disease [2,3,4,5,6]. The progress from simple steatosis to NASH involves the production of reactive oxygen species and reactive nitrogen as well as the increase of lipotoxicity and inflammatory cytokines [7,8]. The main sources of these oxidants are the mitochondria, which are organelles also responsible for fat metabolism. Therefore, mitochondria play an important role in the pathology and progression of steatosis. The classical “two hit theory” is an important pathogenesis model of NAFLD [9,10,11]. First, insulin resistance increases dietary intake and liver fat production, which may lead to the accumulation of triglycerides and free fatty acids in the liver. Secondly, excessive free fatty acids can promote the oxidation of fatty acids in the liver, produce excessive superoxides, impair the function of hepatocyte mitochondria, and result in failure to produce enough adenosine triphosphate (ATP) to ensure normal hepatocyte function, resulting in lipid metabolism regulation, blood glucose, and protein synthesis disorders [12]. Lipid peroxidation, mitochondrial dysfunction, and inflammation may eventually lead to hepatocyte injury and liver fibrosis. However, the “multiple hit theory” is now widely recognized as an accurate model of the pathogenesis of NAFLD, including the interaction of genetic and environmental factors and crosstalk changes between different organs and tissues including adipose tissue, pancreas, gut, and liver [13,14,15,16,17]. Moreover, existing research suggests that NAFLD may be a mitochondrial disease [18].

There are seven sirtuins (SIRT1–7) in mammals that play roles in regulating metabolism in different tissues. SIRT family members are distributed in different cellular compartments and coordinate the cellular response of organisms to stress by promoting mitochondrial oxidation droplet metabolism and inducing stress tolerance. Recent studies have found that SIRT family members play a significant role in regulating hepatic steatosis, NAFLD, and hepatocellular carcinoma, affecting mitochondrial energy metabolism, oxidative stress, and apoptosis by regulating the acetylation status of upstream and downstream proteins [19,20,21]. SIRT3, SIRT4, and SIRT5 are mainly located in the mitochondria and involved in deacetylation [22]. SIRT3, SIRT4, and SIRT5 play an important regulatory role in NALFD by affecting mitochondrial function to regulate fatty acid oxidative metabolism, inflammation levels, and oxidative stress [20,23,24]. Therefore, it is necessary to connect the active state of the SIRT family with the occurrence and development of NAFLD. This study reviews the role of the SIRT family in NAFLD and the molecular proteins involved, in order to further study the pathogenic mechanism and possible treatments of NAFLD.

## 2. The Structure, Function, and Diversity of Sirtuins

Sirtuins (SIRT1–7) all belong to the same category of histone deacetylases (HDACs), and the overall structure of HDAC domains is similar among all isomers. Each isomer has a large Rossmann folding domain for NAD+ binding and a small domain containing zinc-binding band modules [25,26]. For example, SIRT2 consists of a NAD+-bound Rossmann fold and a zinc-binding motif [27]. SIRT5 consists of 14 α helices and 9 β chains and also has a zinc-ion-binding domain and Rossmann folding domain [26], which is similar to other sirtuins. SIRT family members are a kind of NAD+-dependent III deacetylase, which control key cellular processes in the nucleus, cytoplasm, and mitochondria to maintain metabolic balance, reduce cell damage, and inhibit inflammation. SIRT family members have been proven to play important roles in regulating the occurrence and development of NAFLD [28]. SIRT1, the most studied member of the family, is considered to be a metabolic regulator in different tissues and involved in the control of longevity and lipid metabolism, including fatty acid synthesis, oxidation, and adipogenesis [29,30]. SIRT1 deficiency can lead to metabolic disorders such as diabetes, nonalcoholic fatty liver, cardiovascular disease, and neurodegeneration [31,32,33,34]. In addition, SIRT1 shows tumor-inhibitory activity in metabolic-syndrome-related cancers [35,36]. SIRT2 plays a key role in regulating cell cycle, neurodegeneration, tumor inhibition, and inflammation, as well as potential roles in adipocyte differentiation, lipolysis, fat synthesis, and fatty acid oxidation [37,38,39,40,41,42,43,44,45,46,47]. SIRT3, a key regulator of mitochondrial function, is highly expressed in metabolically active tissues and regulates energy production and oxidative stress responses [48,49]. SIRT4 promotes catabolism of branched amino acids and adipogenesis and inhibits insulin secretion and liver lipid oxidation [20,50,51]. SIRT5 inhibits oxidative stress and gluconeogenesis, enhances fatty acid oxidation, and participates in regulating mitochondrial energy metabolism [24,52,53]. SIRT6 promotes fatty acid oxidation, reduces liver triglycerides, cholesterol, reactive oxygen species (ROS), and inflammatory cytokines, and enhances the antioxidant capacity of the liver [54,55]. SIRT7 participates in cell metabolism and stress, promotes mitochondrial biogenesis, and maintains mitochondrial respiration, which is closely related to aging, fatty liver, and tumorigenesis [56,57,58,59]. SIRT family members play important roles in the study of metabolic diseases, especially some diseases regulated by mitochondria, such as hepatic steatosis, NAFLD, and liver fibrosis, and may make it possible to further clarify the pathogenesis of these metabolic diseases.

## 3. The Role of Mitochondria in NAFLD

Mitochondria are involved in a range of physiological processes, including production of ATP, storage of calcium, release of free radicals, oxidation of free fatty acids, and synthesis of cholesterol [60]. Mitochondrial dysfunction has been widely recognized as an important factor in the occurrence and development of NAFLD induced by high-fat diet (HFD) [61,62,63]. HFD can lead to abnormal accumulation of triglycerides and imbalance of liver mitochondrial function [64,65,66,67]. Liver mitochondrial dysfunction, represented by decreased energy production and impaired redox balance, plays a central role in development of the first and second stages of NAFLD [68,69,70]. Mitochondrial dysfunction increases ROS, oxidative stress, and defective bioenergy substances, resulting in liver fat accumulation and injury, which may promote the progression of liver disease from nonalcoholic fatty liver to NASH through the mechanism of liver inflammation, necrosis, and fibrosis [71]. Excessive ROS cause mitochondrial membrane peroxidation and the collapse of mitochondrial membrane potential, which disrupts the synthesis and release of cytochrome C in the mitochondria, resulting in irreversible cell damage or death [18]. In addition, mitochondrial dysfunction disrupts the dynamic balance of fat in hepatocytes and causes fat accumulation, which leads to lipotoxicity [62]. Lipotoxicity is related to the increase of mitochondrial apoptosis, which is characterized by the decrease of mitochondrial potential, the increase of ROS production and cytochrome C infiltration into the nucleus, and the activation of the caspase-9 apoptosis signal [72]. In addition, the activity imbalances of transcription factor steroid regulatory element binding protein-1c (SREBP-1c), peroxisome proliferator-activated receptor-α (PPARα), nuclear factor-kB (NF-kB), and peroxisome proliferator-activated receptor gamma coactivator-1α (PGC-1α) are key regulators of mitochondrial biosynthesis and oxidative phosphorylation function in NAFLD [73,74,75,76,77,78,79,80]. Further study of potential target molecules may allow protection of cells from mitochondrial dysfunction, which will be very important for the treatment of mitochondria-mediated NAFLD [81].

## 4. The Role of the SIRT Family in Mitochondria

There are seven SIRT proteins in mammals, which are located in different subcellular locations and regulate different cellular functions, including physiological, metabolic, and epigenetic regulatory roles [82]. SIRT1, SIRT6, and SIRT7 are nuclear proteins; SIRT2 is localized in the cytoplasm; SIRT3, SIRT4, and SIRT5 are localized in the mitochondria [22]. The SIRT family members have been shown to be involved in regulation of histone acetylation, inflammation, gene transcription, fatty acid and glucose metabolism, insulin regulation and adipocyte maturation, and mitochondrial function and biogenesis [83]. As a key molecule in the regulation of mitochondrial biogenesis, PGC-1a plays an important role in adaptive thermogenesis, mitochondrial formation, and the regulation of glucose and lipid metabolism. SIRT1 can promote the accumulation of PGC-1 α in the nucleus, which leads to gene transcription necessary for mitochondrial function and biogenesis [84]. SIRT2 regulates mitochondrial remodeling and oxidative metabolism through the MEK1-ERK-Drp1 and AKT1-Drp1 signaling pathways [85]. SIRT3 is mainly expressed in mitochondria and has multiple protective effects on mitochondria under oxidative stress, hyperglycemia, fatty acid composition, and myocardial infarction [86,87,88]. SIRT3 has been found to be most associated with various interacting proteins, such as those involved in amino acid metabolism, fatty acid oxidation, the tricarboxylic acid cycle, and the ETC/OXPHOS complex, while SIRT4 and SIRT5 are associated with fewer proteins and may regulate narrower mitochondrial pathways [89,90]. In addition, SIRT3 controls the overall acetylation of mitochondrial proteins, increases the level of cell ATP, and induces mitochondrial biogenesis [48,91]. SIRT3 can also promote mitochondrial oxidative metabolism in response to nutritional stress and membrane depolarization [48,89]. SIRT3 deficiency decreases the level of ATP and increases the level of ROS in cells [92]. On the other hand, the increase of mitochondrial SIRT3 content promotes the expression of mitochondrial coding genes through the AMPK-PGC1α-ERRα signal pathway, which increases ATP synthesis and complex I activity and decreases mitochondrial ROS production [93]. SIRT4 acts as a metabolic regulator between glycolysis and the TCA cycle, not only inhibiting malonyl-CoA carboxylase, which represses fatty acid oxidation and promotes lipid synthesis, but also further inhibiting pyruvate dehydrogenase and stimulating mitochondrial ATP production [94,95]. Inhibition of SIRT4 can promote mitochondrial gene expression and increase fatty acid oxidation [50]. In addition, SIRT4 directly or indirectly regulates a variety of mitochondrial functions closely related to the progression of aging-related diseases such as type 2 diabetes, neurodegeneration, and cancer [95,96,97,98]. As a global regulator of mitochondrial lysine succinylation, SIRT5 participates in the regulation of metabolic networks [99]. SIRT6 protects mitochondria and plays an anti-apoptotic role by activating the AMP-activated protein kinase (AMPK) pathway [100]. SIRT7 regulates mitochondrial biogenesis, ribosomal biosynthesis, and DNA repair through epigenetic mechanisms, and coordinates glucose metabolism to maintain energy homeostasis [58]. In short, SIRT family members regulate mitochondrial function through a variety of mechanisms and participate in various metabolic pathways such as inflammation, endoplasmic reticulum stress, insulin resistance, fatty acid oxidation, steatosis, and so on. These metabolic pathways are also involved in the progress of NAFLD, so it is necessary to further study the role of the SIRT family in NAFLD, especially on the basis of regulation of mitochondrial biogenesis.

## 5. The Role of the SIRT Family in Mitochondrial Biogenesis and NAFLD

### 5.1. SIRT1 and NAFLD

SIRT1 is a post-translational regulator which plays an important role in gene transcription, apoptosis, cell cycle, metabolism, and development. SIRT1 has been shown to regulate longevity and cellular metabolism, including fat metabolism, anti-inflammatory response, insulin secretion, cell differentiation, and senescence; increase cell cycle arrest; improve resistance to oxidative stress; and respond to hunger and calorie restriction [35,101,102,103,104,105,106,107]. The activation of SIRT1 promotes the transcription of genes regulating mitochondrial biogenesis to maintain energy and metabolic homeostasis as well as mediating the deacetylation of downstream target proteins, such as PGC-1a, involved in fatty acid oxidation and mitochondrial function [84,108,109,110]. Related studies have shown that oxidative stress and PGC1α-mediated mitochondrial dysfunction are involved in the pathological changes of NAFLD lipid metabolism caused by SIRT1 silencing in type 2 diabetes mellitus [111]. The activation of SIRT1 can promote SREBP-1c phosphorylation, reduce the level of nuclear mature SREBP-1c, inhibit liver fat synthesis, promote fat decomposition, and increase fatty acid oxidation by regulating the SIRT1/AMPK/SREBP-1c signal pathway [30]. In the HFD mouse model of obesity and NAFLD patients, the expression of SIRT1 is inhibited, resulting in liver metabolic damage [112,113]. At the cellular level, studies have shown that inhibition of SIRT1 can lead to decrease of cell activity, apoptosis, lipid accumulation and production of reactive oxygen species, and dysfunction of mitochondria, while activation of SIRT1 can participate in mitochondrial biogenesis through PGC-1α and participate in the balance of autophagy regulatory proteins [114]. Inhibition of SIRT1-mediated mitochondrial biogenesis may lead to mitochondrial protein stress and unfolded protein response (UPR) against the background of NASH [115]. NAFLD experimental animal model studies show that SIRT1, as a negative regulator of UPR signals, inhibits mechanistic target of rapamycin complex 1 (mTORC1) and endoplasmic reticulum stress, reduces hepatic steatosis, improves insulin resistance, and restores glucose homeostasis [116]. The ATG3-JNK1-SIRT1-CPT1 signaling pathway is also involved in fatty acid metabolism in the liver, which increases the expression of SIRT1 by inhibiting ATG3, thus improving mitochondrial function [21]. Cadmium exposure can cause liver mitochondrial dysfunction and fatty acid oxidation deficiency, and significantly inhibit the liver SIRT1 signal pathway [117]. In addition, patients with NAFLD are easily affected by hepatic ischemia-reperfusion injury. Studies have shown that the RNLS-STAT3-SIRT1 signal pathway can reduce ROS production, improve mitochondrial function, and effectively reduce hepatic ischemia-reperfusion injury [118]. SIRT1 deacetylates members of the fork box O (FoxO) family, affecting the downstream pathway that controls autophagy [84,119]. Specific inhibition of SIRT1 expression can lead to a decrease of FOXO3a expression, then induced autophagy and decreased PGC-1 α expression and mitochondrial dynamics, and then reduced hepatic ischemia-reperfusion injury [120]. SIRT1 is also the main target of many plant flavonoids, including resveratrol, which has beneficial metabolic and anti-aging effects [121]. SIRT1, when activated by these phytochemicals, regulates mitochondrial biogenesis and mitosis mediated by PGC-1α. Silencing SIRT1 eliminates the promoting effect of blueberry juice and probiotics on the expression of PGC-1α and the protective effect of mitochondrial dysfunction in NAFLD rats [122]. The natural product diosgenin downregulates the gene expression of SREBP-1c, FAS, and SCD and upregulates the gene expression of FoxO1, ATGL, and CPT through the SIRT1/AMPK signaling pathway, promoting lipolysis and thus preventing the progression of NAFLD (Appendix A) [123]. In addition, pharmacological stimulation of SIRT1 attenuates hepatic steatosis through PRKA-independent, SIRT1-mediated autophagy and attenuates hepatic ischemia-reperfusion injury by restoring mitochondrial function and enhanced autophagy [29,124].

### 5.2. SIRT2 and NAFLD

SIRT2 is highly abundant in metabolically active tissues, including liver, heart, brain, and adipose tissues, and can deacetylate multiple protein targets [125]. The expression of SIRT2 is significantly decreased in obese mice with fatty liver and NAFLD; liver-specific knockout of SIRT2 aggravates hepatic fat accumulation, inflammation, and insulin resistance induced by HFD, while overexpression of liver SIRT2 improves insulin sensitivity, oxidative stress, and mitochondrial dysfunction in obese mice [19,126]. Studies have shown that SIRT2 deacetylates glucokinase regulatory proteins to restore impaired liver glucose metabolism [127]. SIRT2 negatively regulates adipocyte differentiation by deacetylation of FoxO1, and regulates cell response to oxidative stress by deacetylation of FoxO3a [47,128,129]. In the mouse model fed with HFD, the endoplasmic reticulum and mitochondria are close to each other, which depends on acetylated α-tubulin and NAD+ levels. The decrease of NAD+ in the liver is closely related to fat accumulation, increased oxidative stress, inflammation, and impaired insulin sensitivity in the liver [130,131]. It has been found that the assembly of NOD-like receptor family pyrin domain-containing 3 (NLRP3) and apoptosis-related spot-like proteins containing caspase recruitment domain (ASC) is based on the physical pathway between the endoplasmic reticulum and mitochondria. Endogenous ASC is located in the mitochondria, cytoplasm, and nucleus, while endogenous NLRP3 is mainly located in the endoplasmic reticulum [132,133,134]. Under stimulation by the inducer of NLRP3 inflammatory bodies, the mitochondria approach the endoplasmic reticulum of the perinuclear region, resulting in the coexistence of ASC and NLRP3 on the mitochondria. Pharmacological stimulation to restore SIRT2 expression can reverse the activation of NLRP3 inflammatory bodies and reduce the inflammatory effects induced by palmitic acid or HFD. In addition, SIRT2 is involved in the deacetylation of lysine residues 540, 546, and 554 of adenosine triphosphate-citrate lyase (ACLY) [45]. ACLY is a key lipogenic enzyme that catalyzes the consumption of citric acid to acetyl-CoA. Overexpression of SIRT2 reduces the stability of the ACLY protein, inhibits lipid accumulation in hepatocytes, and reduces hepatic steatosis in mice fed with HFD (Appendix A) [45]. SIRT2 maintains metabolic dynamic balance through glucose and lipid metabolism, insulin sensitivity, and inflammatory regulation, which are key pathological processes related to the occurrence and development of NAFLD.

### 5.3. SIRT3 and NAFLD

SIRT3 is located in the mitochondria, has strong deacetylase activity, and plays a key role in maintaining redox dynamic balance, regulating epigenetics, and lipid metabolism [135,136,137]. Under normal circumstances, SIRT3 exists in the form of long chains located in the nucleus. When the external environment changes, SIRT3 forms a catalytic active short chain, which enters the mitochondria and performs the deacetylation function [138]. In the HFD model, the expression of SIRT3 decreases in liver tissue, and its gene knockout leads to a significant decrease in fatty acid metabolic rate and increased triglyceride accumulation in the liver, while overexpression of SIRT3 protects liver function, reduces liver fibrosis and inflammation, and reduces hepatocyte apoptosis [72,139]. The injury of mitochondrial biosynthesis and antioxidant response aggravates the severity of NAFLD induced by HFD, while pharmacological stimulation of SIRT3 expression can improve mitochondrial function and reduce hepatic steatosis [140]. SIRT3 may upregulate BCL2 Interacting Protein 3 mediated mitochondrial autophagy, reduce mitochondrial damage, and inhibit mitochondrial-dependent hepatocyte apoptosis by activating the ERK-CREB signal pathway [72]. The increase of ROS production may be due to the decrease of SOD2 activity and the hindrance of ROS clearance. The inactivation of mitochondrial SIRT3 leads to SOD2 acetylation, which reduces SOD2 activity [141].

In addition, SIRT3 can deacetylate the lysine group at position 122 of manganese-superoxide dismutase (MnSOD), thus increasing the activity of MnSOD, reducing the production of mitochondrial ROS, and participating in antioxidant stress [142]. However, as a negative regulator of autophagy, overactivation of SIRT3 expression in adipotoxicity caused by saturated fatty acids can lead to MnSOD deacetylation, depletion of intracellular superoxide, inhibition of AMPK, and activation of mTORC1, thereby inhibiting autophagy and aggravating lipotoxicity [23]. This is in contrast to the results of other studies [143,144]. SIRT3 as a key regulator of mitochondrial dysfunction and redox homeostasis in NAFLD needs to be further discussed. However, there is evidence that SIRT3 knockout mice induced by HFD show severe metabolic syndrome [139]. Pharmacological stimulation of SIRT3, PGC1α, nuclear respiratory factor-1, and FoxO3 mRNA expression or activation of the SIRT3-AMPK-PGC1α-ERRα signaling pathway can improve mitochondrial dysfunction and oxidative stress [145,146,147]. FoxO3 is the direct target of SIRT3, and its function is to regulate the expression of multiple genes as a forked transcription factor [148]. SIRT3-mediated FoxO3 deacetylation reduces the level of ROS by upregulating antioxidant enzymes MnSOD and catalase (CAT), thus maintaining the reserve capacity of mitochondria and protecting cells from oxidative damage [149]. Located in the upper reaches of the SIRT3/FoxO3 pathway, miRNA-421 can directly recognize and inhibit the expression of SIRT3/FoxO3 protein, resulting in the decrease of MnSOD and CAT, the downstream targets of SIRT3/FoxO3, and the increase of fat accumulation and triglyceride levels in adipocytes, thus participating in the oxidative damage of NAFLD [150]. Pharmacological stimulation can increase the activity of mitochondrial SIRT3, improve the energy deficiency caused by OXPHOS damage, and improve mitochondrial membrane potential, oxygen consumption, and cellular ATP level [151,152]. Mitochondrial trifunctional protein (MTP) plays a key role in the oxidation of long-chain fatty acids. Overexpression of SIRT3 significantly decreases the acetylation of MTP, increases mitochondrial FAO, and decreases hepatic steatosis, CD68, and serum ALT levels [153]. The inactivation of SIRT3 decreases the activity and gene expression of the SDH subunit and increases the level of succinic acid in liver tissue, but it has no significant effect on the production of mitochondrial ROS [154]. Overexpression of RBP4 promotes the acetylation of long-chain acyl CoA dehydrogenase by inhibiting the expression and activity of SIRT3, and significantly hinders the binding of SIRT3 to long-chain acyl CoA dehydrogenase, thus inducing the deterioration of mitochondrial dysfunction and lipid metabolism (Appendix A) [155]. In short, as a key regulator of mitochondrial biogenesis, SIRT3 plays a very important role in the occurrence and development of NAFLD. Further study on the deacetylation function of SIRT3 in mitochondria and its upstream and downstream molecular targets is of great significance for elucidating the pathogenesis of NAFLD and preparing appropriate drugs to prevent and treat NAFLD.

### 5.4. SIRT4 and NAFLD

SIRT4 is also a key regulator of the mitochondrial metabolic pathway, regulating insulin secretion, mitochondrial ATP production, apoptosis and redox pathway [96]. A previous study found that the expression of SIRT1, SIRT3, SIRT5, and SIRT6 in NAFLD is decreased, while the expression of SIRT4 is upregulated, which may promote the progress of NAFLD [156]. The activity of SIRT4 increases under conditions of adequate nutrition, which inhibits the oxidation of fatty acids and promotes the synthesis and metabolism of fat [157]. Mitochondrial trifunctional protein α subunit (MTPα) is a key enzyme in fatty acid β oxidation. MTPα is acetylated at lysine residues 350, 383, and 406 (MTPα-3K), which promotes its protein stability by antagonizing its ubiquitination on the same three lysines and preventing its subsequent degradation. In NAFLD mouse and human livers, the acetylation levels of mitogen α and mitogen α-3K decrease, while the deacetylation level of SIRT4 increases, resulting in the decrease of MTPα protein stability and aggravation of NAFLD (Appendix A) [158]. In primary hepatocytes with SIRT4 knockout, the expression of mitochondria and fatty-acid-metabolizing enzyme genes increases significantly [159]. However, some studies have shown that the SIRT4/Smad4 axis plays a key role in the formation of liver fibrosis, and the upregulation of SIRT4 may reduce lipid accumulation, inflammation, and fibrosis induced by HFD [160]. In terms of mechanism, SIRT4 may negatively mediate fatty acid oxidation in hepatocytes by inhibiting the transcriptional activity of PPARα [159]. In normal hepatocytes, SIRT1 is recruited to PPARα reactants by interacting with PPARα, which catalyzes the deacetylation of N-acetyllysine of PGC-1α to promote fatty acid oxidation [161]. SIRT4 destroys the interaction between SIRT1 and PPARα and weakens the activation of PPARα transcriptional activity through SIRT1 to inhibit fatty acid oxidation [162]. SIRT4 competes with other sirtuins including SIRT1 for β-NAD+, resulting in a decrease in SIRT1 activity and the effect of SIRT1 on PPAR α and fatty acid oxidative transcriptional activity [163,164].

### 5.5. SIRT5 and NAFLD

SIRT5 is also located in the mitochondria. It not only has NAD+-dependent deacetylase activity, but also specifically removes succinyl, glutaryl, and malonic acid groups from lysine residues [52,99,165,166]. The decreased activity of mitochondrial acyl-CoA synthetase in the liver tissues of SIRT5 knockout mice can cause periportal steatosis [167]. The mRNA of SIRT5 binds to splicing factor 2, which is rich in serine/arginine, to regulate the stability of its mRNA, increase the expression of SIRT5, enhance the expression and enzyme activity of carnitine palmitoyl transferase 1A, inhibit hepatocyte ROS formation, and restore mitochondrial function (Appendix A) [168]. In addition, SIRT5 desuccinylation positively regulates fatty acid oxygen and ketone body production [99,169]. In peroxisomes, which have a fatty acid oxidation pathway similar to that of mitochondria, SIRT5 negatively regulates the key protein acyl-CoA oxidase-1 for fatty acid oxidation, and its deletion inhibits mitochondrial fatty acid oxidation and promotes peroxisomes [52].

### 5.6. SIRT6, SIRT7, and NAFLD

Both SIRT6 and SIRT7 can participate in liver lipid metabolism, endoplasmic reticulum stress, and insulin resistance, and regulate the progression of NAFLD. Decreased SIRT6 protein levels in the liver of obese mice and NAFLD patients, and fat-specific SIRT6 deficiency, increased inflammation and insulin resistance induced by HFD in mice [156,170,171]. SIRT6 overexpression can improve hepatic steatosis, insulin resistance, and inflammation, which regulates hepatic steatosis by interacting with USP10 and inhibiting SIRT6 ubiquitin and degradation [172]. In addition, SIRT6 negatively regulates glycolysis, triglyceride synthesis, and fat metabolism through histone H3 deacetylation, thus promoting the dynamic balance of liver triglycerides [173]. SIRT6-deacetylated X-box binding protein 1 (XBP1) is transactivated in Lys257 and Lys297, and promotes the degradation of XBP1s protein through the ubiquitin-proteasome system, thereby reducing endoplasmic reticulum stress and liver steatosis [174]. SIRT7 is a histone H3 lysine 18 (H3K18) deacetylase that binds to the promoters of a specific group of transcriptional suppressor genes. SIRT7 was shown to be induced under endoplasmic reticulum stress and stabilized on the ribosomal protein promoter through interaction with transcription factor Myc, thus inhibiting gene expression, alleviating endoplasmic reticulum stress, and reversing diet-induced fatty liver in obese mice [57]. In addition, liver SIRT7 positively regulates the protein level of nuclear receptor TR4/TAK1 involved in lipid metabolism, thereby activating TR4 target genes to increase fatty acid uptake and triglyceride synthesis/storage [59]. As deacetylases involved in important metabolic functions, the roles of SIRT6 and SIRT7 in mitochondrial function in patients with NAFLD are not clear. Both SIRT6 and SIRT7 are involved in steatosis and endoplasmic reticulum stress, suggesting that they may have favorable roles in NAFLD, but whether they regulate mitochondrial biogenesis needs further study.

## 6. Conclusions

In obese and NAFLD mice and human patients, the expression of the SIRT family decreases, but the expression of SIRT4 may be increased. As SIRT1–7 are a family of NAD+-dependent protein deacetylases, the result is increased mitochondrial protein acetylation, which may in turn contribute to impaired mitochondrial fuel oxidation and respiration, thus contributing to the vicious cycle of “energy starvation”. Reduced ATP levels are responsible for cell vulnerability to any negative stimuli and increased death with consequent increased fibrosis deposition, thus leading to NAFLD progression to cirrhosis [175]. Overexpression of SIRT family members can improve mitochondrial function and reduce oxidative stress and lipid accumulation through different pathways and mechanisms, thus improving the progress of NAFLD. To date, SIRT1–5 have been proven to be involved in mitochondrial biogenesis in NAFLD patients or animal models, but the roles of SIRT6 and 7 in mitochondria are still unclear and need further study. Mechanically, SIRT1 can participate in the occurrence and development of NAFLD by affecting signal molecules such as PGC-1α and SREBP-1c, FoxO1/3, STAT3, and AMPK, and restore the function of mitochondria by regulating mitochondrial fat oxidation and reducing the production of ROS, thus controlling the progress of NAFLD. Similarly, SIRT2–5 are also involved in the regulation of mitochondrial function through regulation of different metabolic molecules and signal pathways, such as ACLY, SOD2, MnSOD, COX IV, PGC1α, ERRα, FOXO3, CAT, NRF1, SDH, SUCNR1, MTP, FAO, MTPα, CPT1a, SRSF2, and so on. SIRT6 and 7 have been proven to be involved in the regulation of mitochondrial function in other diseases, but in terms of their role, we speculate that they may be involved in the occurrence of NAFLD, so they can also be used as important targets for future research in NAFLD. In short, the SIRT family plays an important role in regulation of mitochondrial biogenesis, oxidative stress, fatty acid metabolism, inflammation, and insulin resistance in NAFLD.

## Data Availability

Not applicable.

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
