# Peer review of "Progress in Nonalcoholic Fatty Liver Disease: SIRT Family Regulates Mitochondrial Biogenesis"

_biomolecules, 2022, doi:10.3390/biom12081079_

Round 1

Reviewer 1 Report

The authors C. Zheng and M. Chen have contributed a well-rounded review on the SIRT family and its involvement in NAFLD development and progression, focusing on the role of mitochondrial molecular pathways.

I would like to comment on the following:

1.      L. 27, “range of liver diseases” needs to be changed to “range of liver disease of progressive severity”. Hepatocellular carcinoma and cirrhosis are in reality consequences of NAFLD and the culmination of the disease, since they can be attributed to other types of liver disease, such as viral hepatitis.

2.      L. 58, “played the function” needs to be changed to “operate as”

3.      In Section 2, the title talks about structure of the SIRT family, however this is not addressed in the text. The structure of most of the SIRTs has been solved by crystallography and there are studies in literature to describe common features of the proteins, such as the Yang et al., Sci China Life Sci. 2017 Mar;60(3):249-256. A few sentences on this subject should be added.

4.      L. 147. Better description of SIRT4-related functions in mitochondria is needed (relevant to Ref. 91-96).

5.      L. 174, “obese mice induced by HFD” do you mean “the HFD mouse model of obesity”?

6.      L. 203. “reduces lipid metabolism” does not make much sense. I think the authors are referring to the fact that diosgenin has been shown to promote lipolysis and attenuate lipogenesis, ie. reduce fatty acid synthesis by down-regulating the relevant transcription factors. The sentence needs to be rewritten for better clarity.

7.      In my opinion there is unwanted extensive use of the past tense in many parts of the text.

Examples include lines 8-14 in the Abstract, lines 34-37 and 50-53 in the Introduction section etc.

Author Response

Dear Reviewer,

Thank you for your kind letter and your careful work regarding our manuscript. We have revised the manuscript in accordance with your comments. And point-by-point responses to the comments were as follows:

1.L. 27, “range of liver diseases” needs to be changed to “range of liver disease of progressive severity”. Hepatocellular carcinoma and cirrhosis are in reality consequences of NAFLD and the culmination of the disease, since they can be attributed to other types of liver disease, such as viral hepatitis.

Response:Thanks. We have replaced “range of liver diseases” with “range of liver disease of progressive severity”.

2.L. 58, “played the function” needs to be changed to “operate as”

Response:Thanks. We have replaced “played the function” with “operate as”.

3.In Section 2, the title talks about structure of the SIRT family, however this is not addressed in the text. The structure of most of the SIRTs has been solved by crystallography and there are studies in literature to describe common features of the proteins, such as the Yang et al., Sci China Life Sci. 2017 Mar;60(3):249-256. A few sentences on this subject should be added.

Response:3. We have supplemented information. sirtuins (SIRT1-7) belong to the same category in histone deacetylases (HDACs) classification, and the overall structure of HDACs domains of all isomers is similar. Each isomer has a large Rossmann folding domain for NAD+ binding and a small domain containing zinc binding band modules[25, 26]. For example, SIRT2 consists of a NAD+-bound Rossmann fold and a zinc-binding motif[27]. SIRT5 consists of 14 α helices and 9 β chains and also has zinc ion binding domain and Rossmann folding domain[26], which is similar to other sirtuins. Please see page 2, line 65-71.

4.L. 147. Better description of SIRT4-related functions in mitochondria is needed (relevant to Ref. 91-96).

Response:Thank you for your suggestion. We have re-written this sentence.SIRT4 acts as a metabolic regulator between glycolysis and TCA cycle, not only inhibiting malonyl-CoA carboxylase, where represses fatty acid oxidation and promotes lipid synthesis, but also further inhibiting pyruvate dehydrogenase and stimulating mitochondrial ATP production[94, 95]. Inhibition of SIRT4 could promote mitochondrial gene expression and increase fatty acid oxidation[50]. In addition, SIRT4 directly or indirectly regulates a variety of mitochondrial functions closely related to the progression of aging-related diseases, such as type 2 diabetes, neurodegeneration and cancer[95-98].Please see page 3, line 148-156.

5.L. 174, “obese mice induced by HFD” do you mean “the HFD mouse model of obesity”?

Response:Yes, we have replaced “obese mice induced by HFD” with “the HFD mouse model of obesity”.

6.L. 203. “reduces lipid metabolism” does not make much sense. I think the authors are referring to the fact that diosgenin has been shown to promote lipolysis and attenuate lipogenesis, ie. reduce fatty acid synthesis by down-regulating the relevant transcription factors. The sentence needs to be rewritten for better clarity.

Response:Thank you for this valuable comment. We have corrected the sentence.The natural product diosgenin downregulates the gene expression of SREBP-1c, FAS and SCD and upregulates the gene expression of FoxO1, ATGL and CPT through SIRT1/AMPK signaling pathway, promoting lipolysis and thus preventing the progression of NAFLD[123]. Please see page 5, line 209-212.

7.In my opinion there is unwanted extensive use of the past tense in many parts of the text.Examples include lines 8-14 in the Abstract, lines 34-37 and 50-53 in the Introduction section etc.

Response:Thank you for your kind suggestion. We have actively sought the help of English professionals.This manuscript has been modified by English-speaking professional.

Special thanks to you for your good comments.

Mingkai Chen

Reviewer 2 Report

I read with great interest the paper “Progress in nonalcoholic fatty liver disease: SIRT family regulates mitochondrial biogenesis" by Zeng et al.

The article is well written. Paper design is fine. The article is logically divided into sections and subsections.

Comment:

1.      Conclusion: The authors reported that in obese and NAFLD mice and patients, the expression of SIRT family decreased. As Sirtuins (SIRT1–SIRT7) are a family of NAD+-dependent protein deacetylases, the result is an increased mitochondrial protein acetylation, which may in turn contribute to impaired mitochondrial fuel oxidation and respiration, thus contributing to the vicious cycle of “energy starvation”. Reduced ATP levels is responsible for cell vulnerability to any negative stimuli and increased death, with consequent increased fibrosis deposition, thus leading to NAFLD progression to cirrhosis (doi: 10.3390/antiox10020270)

Author Response

Dear Reviewer,

On behalf of my co-authors, we thank you very much for giving us an opportunity to revise our manuscript, especially your positive and constructive comments and suggestions on our manuscript entitled “Progress in nonalcoholic fatty liver disease: SIRT family regulates mitochondrial biogenesis”.

The main corrections in the paper and the responds to the your comments are as flowing: 

1) Response to comment:

1. Conclusion: The authors reported that in obese and NAFLD mice and patients, the expression of SIRT family decreased. As Sirtuins (SIRT1–SIRT7) are a family of NAD+-dependent protein deacetylases, the result is an increased mitochondrial protein acetylation, which may in turn contribute to impaired mitochondrial fuel oxidation and respiration, thus contributing to the vicious cycle of “energy starvation”. Reduced ATP levels is responsible for cell vulnerability to any negative stimuli and increased death, with consequent increased fibrosis deposition, thus leading to NAFLD progression to cirrhosis (doi: 10.3390/antiox10020270)

Response: We have re-written this part according to your suggestions. And we have added the cite reference in conclusion(line 366-372, page 8).

Special thanks to you for your good comments.

Mingkai Chen
